Bacterial contamination of Nigerian currency notes: A comparative analysis of different denominations recovered from local food vendors

http://orcid.org/0000-0002-0835-5872 Ofoedu Chigozie E. 1 chigozie.ofoedu@futo.edu.ng
Iwouno Jude O. 1
Agunwah Ijeoma M. 1
Obodoechi Perpetual Z. 1
http://orcid.org/0000-0003-4475-8887 Okpala Charles Odilichukwu R. 2 charlesokpala@gmail.com
Korzeniowska Małgorzata 2
1 Department of Food Science and Technology, School of Engineering and Engineering Technology, Federal University of Technology , Owerri, Imo State , Nigeria
2 Department of Functional Food Products Development, Faculty of Biotechnology and Food Science, Wroclaw University of Environmental and Life Sciences , Wroclaw , Poland
Grohmann Elisabeth
Electronic publication date: 2021 Jan 19
Publication date: 2021
Volume: 9
Electronic Location ID: e10795
Received 2020 Oct 5; Accepted 2020 Dec 28
Copyright: © 2021 Ofoedu et al.
Copyright year: 2021
Copyright holder: Ofoedu et al.
License: This is an open access article distributed under the terms of the Creative Commons Attribution License, which permits unrestricted use, distribution, reproduction and adaptation in any medium and for any purpose provided that it is properly attributed. For attribution, the original author(s), title, publication source (PeerJ) and either DOI or URL of the article must be cited.
License URL: https://creativecommons.org/licenses/by/4.0/

Keywords: Microbial contamination, Naira notes, Food handlers, Escherichia coli, Staphylococcus spp., Klebsiella spp.

Funding: Wroclaw University of Environmental and Life Sciences, Poland This work was supported by the Wroclaw University of Environmental and Life Sciences, Poland. The funders had no role in study design, data collection and analysis, decision to publish, or preparation of the manuscript.

==============================
Microbial transmission, on the surface of any currency note, can either be through direct (hand-to-hand contact) or indirect (food or other inanimate objects) means. To ascertain the degree of bacterial load enumerated during the handling of money and food items, particularly on currency note by denominations, should be of public health importance. Despite the available literature regarding microbial contamination of Nigerian currency notes, there is still paucity of information about how microbial contamination/load differ across the denominations specific to different food vendors. In this context, therefore, the current study investigated bacterial contamination of Nigerian currency notes via a comparative study of different denominations (₦1,000, ₦500, ₦200, ₦100, ₦50, ₦20, and 10, and ₦5) recovered from local food vendors. Specifically, the different food handlers/vendors included fruit, meat, vegetable, fish, and grain/cereal sellers. All emergent data from 8 × 5 factorial design of experiment were of duplicate measurements. To consider the currency denominations and food vendor type, a one-factor-at-a-time analysis of variance (ANOVA) was conducted. Results showed that about 81.7% of currency notes were contaminated with either Escherichia coli, Klebsiella spp. or Staphylococcus spp. in varying degrees. The higher denominations of ₦500, ₦200, and ₦100 note, with the exception of ₦1,000 note, recorded increased degree of contamination over the lower denominations of ₦50, ₦20, ₦10, and ₦5 note. Based on the total viable count (TVC), the ₦100 currency note appeared the most contaminated (1.32 × 105 cfu/ml) whereas ₦5 note appeared the least contaminated (1.46 × 104 cfu/ml). The frequency of isolated bacteria on currency notes from vegetable, meat, and fish sellers were significantly higher (p < 0.05) compared to other food vendors. The degree of bacterial contamination of the current work appears chiefly dependent on the food vendor type and currency denomination(s). This work calls for increased awareness and education among food vendors and ready-to-eat food sellers. Doing this would help mitigate the possible cross-contamination between currency notes and foodstuff. Through this, consumers would know more about the potential health risks such simultaneous activities (of handling currency notes and foodstuff) do pose on food safety.

Introduction

Worldwide, currency notes and money in general serve as means of economic exchange of goods and services, to defer payments and settle debts (Awe et al., 2010; Ogunleye, 2005; Okon et al., 2003). Between the late 1800s and early 1900s, scientists postulated the association of handling money with disease transmission. Subsequently, by modern scientific techniques, these postulations confirmed that pathogenic organisms can be isolated from currency/money surfaces (Alemu, 2014; Awe et al., 2010). For example, Citrobacter spp., Escherichia coli, Mycobacterium spp., Pseudomonas aeroginosa, Salmonella spp., and Staphylococcus aureus, are among the examples of foodborne pathogenic microorganisms reported on currency notes (Awe et al., 2010). By adhering to various surfaces, food pathogens such as E. coli, S. aureus and Salmonella spp. could remain viable for hours or even days of post-contamination (Okpala & Ezeonu, 2019). However, whether it is between clean and dirty hands, the movement of currency notes especially within the agrofood supply chain would never stop. This inevitable situation potentially facilitates continued occurrence of microbial contamination and proliferation between currency notes and foodstuffs even more likely (Agarwal et al., 2015; Thiruvengadam et al., 2014). To reiterate, the process of microbial contamination and more importantly, its subsequent transmission, the latter with respect to the surface of any currency note, has been understood to be of either direct (hand-to-hand contact) or indirect (food or other inanimate objects) means (Cooper, 1999).

Even though consumers can help to prevent foodborne disease incidence, the different sources from which microorganisms are able to transfer to food is not new. For instance, microbial contamination takes place during the various stages of food preparation. Another instance, fruits on trees and vegetables grown on the soil are naturally microbiologically contaminated. Some cells of such microbes could still remain even after washing (Okpala & Ezeonu, 2019). Besides foodstuffs as well as drinking water that could get contaminated, there remains a wide spectrum of microbial pathogens that can contaminate animals and food products, all of which are among the fundamental causes of foodborne disease incidence and spread (Okpala et al., in press). During the food handling processes within the agrofood supply chain, the contamination of currency notes can take place, particularly involving diverse flora and fauna, aerosols generated by coughing and sneezing, anal region, wounds, to the skin, water, and soil (Agarwal et al., 2015; Thiruvengadam et al., 2014). Currency notes, even before it would reach the bank and in the process of circulating and passing through hands during daily transactions, can equally transmit the pathogenic microbes (Awodi & Nock, 2001; Yakubu, Ehiowemwenguan & Inetianbor, 2014). Besides the large surface area of any given currency note, a number of pathogenic microorganisms, not only capable of surviving on these notes but also, can serve as useful candidates of foodborne pathogens (Michaels, 2002; Podhajny, 2004) and can increase the probability of foodborne disease incidence/spread. The latter can also serve as a useful indicator of poor environmental hygiene and sanitation levels, all of which remains of great public health importance (Cooper, 1999).

Relevant literature about microbial status and survival of pathogens on currency notes have been shown by many workers in Turkey, the United States, Australia, India, Egypt, and China (Xu, Moore & Millar, 2005; Goktas & Oktary, 1992; Pope et al., 2002; Food Science Australia, 2000; El-Dars & Hassan, 2005; Singh, Thakur & Kalpana, 2002). Other studies regarding contamination ascribed to microbial load specific to national currency notes have been reported in Bangladesh (Ahmed et al., 2010; Hosen et al., 2006), Ethiopia (Alemayehu & Ashenafi, 2019), India (Rote, Deogade & Kawale, 2010), Iran (Dehghani, Dehghani & Estakhr, 2011), Nepal (Lamichhane et al., 2009; Prasai, Yami & Joshi, 2008), Nigeria (Awe et al., 2010; Kawo et al., 2009; Oyero & Emikwe, 2007; Umeh, Juluku & Ichor, 2007), Saudi Arabia (Ghamdi et al., 2011; Rashed et al., 2006), South Africa (Igumbor et al., 2007), as well as Sudan (Saadabi et al., 2010). In Europe, Mändar et al. (2016) studied microbial contamination of euro money, whereas in the USA, Michaels (2002) reported on handling money and serving ready-to-eat food, which considered the same gloved hands or without hygiene intervention, and provided in food service establishments, would introduce the risk of cross-contamination to foods. In the global front, Vriesekoop et al. (2010) performed the hygiene status of some world’s currencies by capturing food outlets in 10 different countries (Australia, Burkina Faso, China, Ireland, Netherlands, New Zealand, Nigeria, Mexico, the United Kingdom, and the United States). By assessing the public health risks associated with the simultaneous handling of food and money, Brady & Kelly (2000) showed that coagulase-positive Staphylococci could be present on the currency note surfaces. The environment remains a critical player in food-related microbial transmission process to humans. The environment would also compose materials that are viable candidates for the (microbial) pathogens (Anderson & May, 1991; Struthers & Westran, 2003).

Besides currency note contamination with bacteria that bring about wide range of diseases (Pope et al., 2002), how it is able to exchange through hands especially within the food supply chain (Agarwal et al., 2015; Thiruvengadam et al., 2014), together with the poor sanitation practices that could arise in the market, restaurants, and slaughterhouses is likely to reflect how multi-resistant microbial strains are able to cross-contaminate (Emikpe & Oyero, 2011; Oyero & Emikwe, 2007). Despite the available literature about microbial contamination of Nigerian currency note (Emikpe & Oyero, 2011; Enemour, Victor & Oguntibeju, 2012; Oyero & Emikwe, 2007; Umeh, Juluku & Ichor, 2007; Uneke & Ogbu, 2007), relevant information regarding how microbial contamination/load differ across the denominations is still insufficient. It is reasonable to say that food handlers in Nigeria, oftentimes, after handling currency notes, do fail to properly wash or sanitise their hands and other food/food-related facilities. To better understand how (pathogenic) microorganisms get enumerated, and subsequently circulate between foodstuffs and different currency denominations/notes, should be of consumer health concern. This should be considered particularly important with respect to food handlers in Nigeria who many a times have to perform financial duties alongside foodstuffs. For the reason that foodstuffs would most likely differ with microbial contaminants/load, however, such additional knowledge and understanding on microbial enumeration as well as circulation between foodstuffs and different currency denominations/notes could help, not only in identifying the actual sources of the foodborne diseases, but also, in enlightening the food handlers, food traders, health workers, and the general public as a whole, about the inherent (public) health risks potentially associated with the currency notes, specifically when not handled in a hygienic safe manner. In this context, therefore, the current study investigated the bacterial contamination of Nigerian currency notes via a comparative study of different denomination notes recovered from local food vendors.

Materials and Methods

Schematic overview of experimental programme

The schematic overview of the experimental programme, from the collection of currency note samples, preparation of currency note samples for microbial determinations, inoculation, incubation, to identification of different bacterial isolates, is shown in Fig. 1. To reiterate, this current study was directed to investigate the bacterial contamination of Nigerian currency notes via a comparative study of different denomination notes recovered from local food vendors. Specifically, the different food handlers/vendors included fruit, meat, vegetable, fish and grain/cereal sellers. The different denominations included ₦1,000, ₦500, ₦200, ₦100, ₦50, ₦20, and ₦5 currency notes. For the reason that there were 5 different food handler/vendor types, and 8 different currency note denominations, a factorial design of experiment was deemed appropriate for this study, specifically 8 × 5 factorial type.

Figure 1 The schematic overview of the experimental programme, from the collection of currency note/paper samples, preparation of currency note samples for microbial determinations, inoculation, incubation, to identification of different bacterial isolate.

Collection currency notes

A total of 80 samples of Nigerian currency notes consisting of 10 pieces of eight (8) different denominations (₦1,000, ₦500, ₦200, ₦100, ₦50, ₦20, ₦10, and ₦5) were randomly collected from different food vendor types at Relief Market in Owerri, Imo State, Nigeria. This Relief Market is among major markets that provide the food demands of the ever increasing Owerri population, projected excess of 800,000 as of 2019 (MacroTrends, 2020). Each currency denomination was aseptically collected from food vendors and placed in an ultra violet (UV) sterilized polyethylene bag and shortly after, transported to Food Microbiology laboratory of the Department of Food Science and Technology, Federal University of Technology, Owerri for bacteriological analysis. At all instances, the time period from point of collection of currency note, to arrival of laboratory did not exceed 2 h.

Preparation of materials/samples

Ringer’s solution (quarter-strength) was prepared, by dissolving one ringer’s solution tablet (Merck, Darmstadt, Germany) with a composition of 2.25 g/L sodium chloride, 0.015 g/L potassium chloride, 0.12 g/L calcium chloride, and 0.05 g/L sodium bicarbonate, in 500 mL of distilled water. All media, diluents, glasswares, and forceps used were sterilised by autoclaving at 121 °C for 15 min at 15 psi, while the wire loops were sterilised by flaming. Materials were allowed to cool, wrapped in an aluminum foil, and stored until needed for microbiological analysis.

Upon arrival at the laboratory after sample collection, each Naira note was aseptically inserted into a beaker containing 100 mL of sterile ringer’s solution, and allowed to stand for 30 min at ambient temperature (25–28 °C). During this 30 min period, the beaker was gently and repeatedly shaken as it is widely believed to facilitate the detachment of the adhered microbes (bacteria) from the Naira currency surface as much as possible into the solution. Subsequently, the Naira note was aseptically removed from the beaker using sterile forceps. Thus, the beaker content (washed liquor of soaked notes) served as the resultant test sample for bacterial inoculation, so as to determine the (bacterial) load as well as enumerate the type. The prepared suspensions for test inoculation were used within 2 h to avoid the risk of cross contamination from the environment.

Serial dilution, incubation and counting of bacterial colonies

Serial dilution (10−2 and 10−3) of the washed liquor of soaked notes was performed. Subsequently, aliquot sample (0.1 ml) of each was inoculated on nutrient agar plates using spread plate method, and incubated at 37 °C for 24 h. This was performed in duplicates for each (Naira) currency denomination note per food vendor type. To verify testing conditions, a negative control (with no test material included) was performed using the chosen diluent, inoculated on nutrient agar plates and incubated at 37 °C for 24 h. The control served strictly for the validity and verification of the testing conditions.

The colony counter (Astor 20 Colony Counter, Astori Tecnica, Italy) was used to determine the colony numbers on different plates. The arithmetic mean of the counts per medium were recorded, and resultant colony forming units per millilitre (cfu/ml) in the original inoculum, was determined consistent with methods described previously (Cheesbrough, 2000).

Bacterial characterisation and identification

The characteristic identification (and not quantification) of bacterial colonies were based on its morphology and Gram reaction, catalase and coagulase tests; biochemical tests for indole production, citrate utilization, and urase activity; triple sugar iron (TSI) agar tests (for glucose, lactose and sucrose fermentation); as well as hydrogen sulfide/gas production tests, and oxidase tests, following the methods described previously (Buller, 2014; Cheesbrough, 2000).

Statistical analysis

All emergent data from 8 × 5 factorial design of experiment as shown in Table 1 were of duplicate measurements. In order to determine the degree of bacterial load that had proliferated on currency note by food vendor type, a one-factor-at-a-time ANOVA was applied. The Fisher’s least significant difference (LSD) was used to resolve mean differences. The statistical significant difference was set at the 95% (p < 0.05) confidence level. The IBM SPSS software version 20 (IBM Corporation, New York, NY, USA) was used to do the data analysis.

Table 1 Factorial experimental design table.

	Factor A	Factor B	
Name	Currency denominations	Food vendor type	
Type	Ordinal	Nominal	
Levels	8	5	
1	₦1,000	Fruit sellers	
2	₦500	Meat sellers	
3	₦200	Vegetable sellers	
4	₦100	Fish sellers	
5	₦50	Grain sellers	
6	₦20	–	
7	₦10	–	
8	₦5	–	

Results

Table 2 shows the total viable count (TVC) of Naira denominations isolated from different food vendors. The TVC on Naira denominations handled by fruit, meat, vegetable, fish and grain sellers respectively ranged from 3.0 × 103 to 2.2 × 105 cfu/ml, from 3.0 × 103 to 1.9 × 105 cfu/ml, from 3.0 × 102 to 1.3 × 105 cfu/ml, from 1.3 × 104 to 2.9 × 105 cfu/ml, and from 5.0 × 102 to 5.1 × 104 cfu/ml. Additionally, the TVC significantly differed (p < 0.05) across different currency notes, which ranged from 1.46 × 104 to 1.33 × 105 cfu/ml, and across different food vendors, which ranged from 8.91 × 103 to 1.18 × 105 cfu/ml Additionally, the TVC trend across the different currency denominations followed: ₦100 (132.72 × 103 cfu/ml) > ₦200 (75.40 × 103 cfu/ml) > ₦500 (66.80 × 103 cfu/ml) > ₦20 (50.90 × 103 cfu/ml) > ₦1,000 (42.26 × 103 cfu/ml) > ₦10 (37.40 × 103 cfu/ml) > ₦5 (14.64 × 103 cfu/ml).

Table 2 Total viable count (cfu/ml) of Naira denominations recovered from different food vendors.

Currency denominations	Food vendor type		
Fruit sellers	Meat sellers	Vegetable sellers	Fish sellers	Grain sellers	Mean × 103	
₦1,000	1.3 × 105	3.9 × 104	3.0 × 102	3.0 × 104	1.2 × 104	42.26	
₦500	9.0 × 103	7.5 × 104	1.3 × 105	1.2 × 105	NGD	66.80	
₦200	3.0 × 104	3.0 × 103	3.0 × 103	2.9 × 105	5.1 × 104	75.40	
₦100	2.2 × 105	1.4 × 105	5.0 × 104	2.5 × 105	3.6 × 103	132.72	
₦50	8 .0 × 103	1.6 × 105	7.0 × 103	1.4 × 105	NGD	63.00	
₦20	1.4 × 104	1.9 × 105	2.0 × 104	3.0 × 104	5.0 × 102	50.90	
₦10	3.0 × 103	8.6 × 104	2.8 × 104	7.0 × 104	NGD	37.40	
₦5	NGD	2.9 × 104	2.7 × 104	1.3 × 104	4.2 × 103	14.64	
Mean × 103	51.75	90.25	33.16	117.88	8.91		
Notes:

Values are the means of duplicate determinations.

NGD, No Growth Detected.

Tables 3–5 respectively show the Escherichia coli, Klebsiella spp. and Staphylococci spp. of Naira denominations recovered from different food vendors. Specifically, the E. coli count isolated from Naira denominations recovered from fruit, meat, vegetable, fish, and grain sellers respectively ranged from 9.0 × 102 to 4.0 × 103 cfu/ml, from 7.0 × 101 to 6.5 × 103 cfu/ml, from 2.0 × 102 to 6.0 × 102 cfu/ml, from 1.0 × 102 to 1.0 × 104 cfu/ml, and from 2.0 × 101 to 8.2 × 102 cfu/ml. Specifically also, the Klebsiella spp. count isolated from Naira denominations recovered from fruit, meat, vegetable, fish, and grain sellers respectively ranged from 2.0 × 102 to 8.5 × 102 cfu/ml, from 4.0 × 101 to 5.2 × 103 cfu/ml, from 1.0 × 102 to 1.8 × 103 cfu/ml, from 1.9 × 102 to 5.0 × 103 cfu/ml, and from 4.0 × 101 to 1.1 × 102 cfu/ml. Specifically also, the Staphylococcus spp. count isolated from Naira denominations recovered from fruit, meat, vegetable, fish and grain sellers respectively ranged from 3.0 × 102 to 3.7 × 103 cfu/ml, from 1.1 × 102 to 3.7 × 103 cfu/ml, from 7.0 × 101 to 6.0 × 103 cfu/ml, from 1.5 × 102 to 6.0 × 103 cfu/ml, and from 4.0 × 101 to 1.6 × 103 cfu/ml.

Table 3 Escherichia coli count (cfu/ml) of Naira denominations recovered from different food vendors.

Currency denominations	Food vendor type		
Fruit sellers	Meat sellers	Vegetable sellers	Fish sellers	Grain sellers	Mean × 103	
₦1,000	NGD	1.2 × 102	4.0 × 102	2.5 × 102	NGD	0.15	
₦500	4.0 × 103	7.0 × 101	2.0 × 102	3.0 × 103	2.0 × 101	1.46	
₦200	9.0 × 102	6.1 × 103	6.0 × 102	5.2 × 102	1.0 × 102	1.64	
₦100	NGD	6.5 × 103	3.0 × 102	1.0 × 104	2.0 × 101	3.36	
₦50	1.2 × 103	2.0 × 103	4.0 × 102	3.2 × 103	8.0 × 102	1.52	
₦20	NGD	5.0 × 103	2.0 × 102	3.0 × 102	NGD	1.10	
₦10	1.3 × 103	3.0 × 102	2.0 × 102	1.0 × 102	4.0 × 101	0.39	
₦5	7.0 × 102	2.0 × 102	2.0 × 102	1.6 × 102	2.0 × 101	0.24	
Mean × 103	1.01	2.54	0.31	2.18	0.12		
Notes:

Values are the means of duplicate determinations.

NGD, No Growth Detected.

Table 4 Klebsiella sp. count (cfu/ml) of Naira denominations recovered from different food vendors.

Currency denominations	Food vendor type		
Fruit sellers	Meat sellers	Vegetable sellers	Fish sellers	Grain sellers	Mean × 103	
₦1,000	NGD	NGD	7.0 × 102	NGD	NGD	0.14	
₦500	8.5 × 102	5.0 × 102	1.8 × 103	5.0 × 103	4.0 × 101	1.64	
₦200	6.0 × 102	5.2 × 103	4.0 × 102	8.5 × 102	NGD	1.41	
₦100	2.7 × 103	NGD	1.4 × 102	NGD	NGD	0.57	
₦50	3.5 × 102	8.0 × 102	1.2 × 103	2.0 × 102	1.1 × 102	0.53	
₦20	7.0 × 102	3.0 × 102	5.0 × 102	1.9 × 102	4.0 × 101	0.35	
₦10	2.0 × 102	4.0 × 101	1.0 × 102	2.0 × 102	6.0 × 101	0.12	
₦5	3.2 × 102	NGD	3.0 × 102	NGD	NGD	0.12	
Mean × 103	0.72	0.86	0.64	0.81	0.03		
Notes:

Values are the means of duplicate determinations.

NGD, No Growth Detected.

Table 5 Staphylococci sp. count (cfu/ml) of Naira denominations recovered from different food vendors.

Currency denominations	Food vendor type		
Fruit sellers	Meat sellers	Vegetable sellers	Fish sellers	Grain sellers	Mean × 103	
₦1,000	3.0 × 102	2.8 × 103	NGD	1.5 × 102	NGD	0.65	
₦500	3.7 × 103	1.6 × 102	7.0 × 101	4.3 × 103	NGD	1.65	
₦200	4.2 × 102	3.7 × 103	6.0 × 103	1.5 × 102	9.2 × 102	2.24	
₦100	8.0 × 102	2.7 × 103	2.0 × 102	4.0 × 103	1.6 × 103	1.86	
₦50	1.9 × 103	1.8 × 102	1.2 × 102	3.5 × 103	5.0 × 101	1.15	
₦20	NGD	4.0 × 102	1.2 × 102	6.0 × 103	4.0 × 101	1.24	
₦10	4.1 × 102	8.0 × 102	9.0 × 102	3.0 × 102	5.0 × 101	0.35	
₦5	NGD	1.1 × 102	NGD	4.0 × 102	1.4 × 102	0.13	
Mean × 103	0.94	1.22	0.93	2.35	0.35		
Notes:

Values are the means of duplicate determinations.

NGD, No Growth Detected.

The mean values of E. coli, Klebsiella spp., and Staphylococci spp. counts isolated from Naira denominations, with emphasis on the different food vendors, is shown in Fig. 2. The results indicate that, specific to E. coli, Klebsiella spp., and Staphylococci spp., the Naira denominations recovered from grain sellers obtained less bacterial contaminants, whereas those recovered from meat and fish sellers obtained more bacterial contaminants. Mean values of E. coli, Klebsiella spp. and Staphylococci spp. counts isolated, with emphasis on the Naira denominations recovered from different food vendors, is shown in Fig. 3. The results indicate that the ₦100 and ₦200 specifically obtained higher E. coli and Staphylococci spp. respectively, whereas ₦500 obtained higher Klebsiella spp. and Staphylococci spp., although not significantly different (p > 0.05). However, the Naira denominations with the least isolated bacterial contaminants appears to be ₦10 and ₦5 (specific to Klebsiella spp.), ₦5 (specific to Staphylococci spp.), and ₦1,000 (specific to E. coli).

Figure 2 Mean values of E. coli, Klebsiella and Staphylococcus count isolated from Naira denominations, with emphasis on the different food vendors.

Figure 3 Mean values of E. coli, Klebsiella and Staphylococcus count isolated, with emphasis on the Naira denominations recovered from different food vendors.

The prevalence of E. coli, Klebsiella spp. and Staphylococci spp. on Naira denominations recovered from different food vendors is shown in Table 6. Regardless of food vendors, 81.7% of Naira denominations recovered had been contaminated by at least one of the determined bacteria, and in the following prevalence trend: E. coli (87.5%) > Staphylococci spp. (85%) > Klebsiella spp. (72.5%). Besides, the total prevalence of bacterial load of Naira denominations, specific to how it was recovered from food vendors, followed this trend: vegetable sellers (91.7%) > meat sellers (87.5%) > fish sellers (87.5%) > fruit sellers (75%) > grain sellers (66.7%).

Table 6 Prevalence of E. coli, Klebsiella sp., Staphylococci sp. on currency notes from different food vendors.

Isolated bacteria	Fruit sellers	Meat sellers	Vegetable sellers	Fish sellers	Grain sellers	Total	
E. coli	5 (62.5%)	8 (100%)	8 (100%)	8 (100%)	6 (75%)	35 (87.5%)	
Klebsiella sp.	7 (87.5%)	5 (62.5%)	8 (100%)	5 (62.5%)	4 (50%)	29 (72.5%)	
Staphylococci sp.	6 (75%)	8 (100%)	6 (75%)	8 (100%)	6 (75%)	34 (85%)	
Total	18 (75%)	21 (87.5%)	22 (91.7%)	21 (87.5%)	16 (66.7%)	98 (81.7%)	

Discussion

The TVC on Naira denominations clearly obtained varying ranges across food vendors (Refer to Table 2). The ₦100 currency denomination obtained the highest TVC (1.32 × 105 cfu/ml), which is consistent with the report of Kawo et al. (2009), attributable to its higher frequency of high usage in today’s Nigeria and across her society’s daily transactions (Adamu, Jairus & Ameh, 2012; Umeh, Juluku & Ichor, 2007). On the other hand, the ₦5 currency denomination obtained the least TVC (1.46 × 104 cfu/ml), probably attributable to its limited use, considering that in recent times, it is actually hard to find any commodities sold for ₦5 in Nigeria. Overall, about 95% of higher denominations (₦1,000, ₦500, ₦200, and ₦100) appeared more contaminated compared to 85% of lower denominations (₦50, ₦20, ₦10 and ₦5), resembling the observation reported by previous workers (Adamu, Jairus & Ameh, 2012).

The fact that E. coli could be isolated, as shown in Table 3, from low level of 2.0 × 101 cfu/ml as found in ₦500, ₦100, and ₦5 denominations of grain sellers, to as high level of 1.0 × 104 cfu/ml as found in ₦100 denomination of fish sellers, might be suggestive of the poor hygienic practices and sanitary condition exercised by these different food vendors, particularly in the handling of the Nigerian currency notes. It is possible that bacterial coliforms could find its way to the surface of the currency notes through other means apart from the food stuffs. For the currency notes to pass through the diverse environments could make it emerge a reservoir, capturing various bacteria including pathogenic E. coli, considered capable of surviving several days on inert surfaces (Pomperayer & Gaylarde, 2000). In addition, the currency notes if poorly handled may result in contamination of foodstuffs (and ready-to-eat foods), unless good hygienic practices (GHPs) are exercised (Barro et al., 2006). Nonetheless, the E. coli remains an important member of Enterobacteriaceace known to bring about food infections and poisoning (Awe et al., 2010). Some E. coli strains can be associated with heat stable enterotoxin production (WHO, 1984; Jensen, Wright & Robison, 1997).

As shown in Table 4, the fact that Klebsiella spp. could also be isolated from Naira denominations from food vendors, from low level of 4.0 × 101 cfu/ml as found in ₦500 and ₦20 denominations of the grain sellers, to high level of 5.2 × 103 cfu/ml as found in ₦200 denomination of the meat sellers, should not be too surprising. This is because this specific bacteria is also an Enterobacteriaceace member. Just like E. coli, Klebsiella spp. can contaminate the water used in moistening the fingers while counting money or cross-contamination from offals. As a rod shaped Gram-negative bacteria, Klebsiella spp. could be found on the skin, mouth, intestinal lining, and if increasingly proliferated, it is likely to bring about urinary tract infections (Prescott, Harley & Klein, 2008). High rates of Klebseilla spp. on some currency notes is not new, as it has been reported in the United States coins and dollar bills (Gadsby, 1998). However, the degree to which currency notes get contaminated with Klebseilla spp. should not be underestimated. Thus, the bacterial contamination of currency notes are not only confined to developing nations.

Staphylococcus spp., a gram-positive bacteria of spherical shape is among common contaminants isolated from currency notes (Xu, Moore & Millar, 2005). In this current study, the rate at which Staphylococcus spp. appeared on the Naira denominations, could be attributed to such factors as contamination between normal skin (hands, fingers, faces) flora, nasal discharge, soil as well as its ubiquitous distribution in the environment (Igumbor et al., 2007; Kumar et al., 2009; Larkin et al., 2009). Additionally, the rubbing off or maybe surfing from a skin flake could facilitate the occurrence of Staphylococcus spp. on the currency notes (Ahmed et al., 2010). Among Staphylococcus spp., for example, S. aureus has the capacity to secrete toxins such as pyrogenic toxin and super antigens, which can bring about health issues like food poisoning as well as toxic shock syndrome (Ayopo, 2010). Given that S. aureus could flourish in the human nose, throat, and skin, the recontamination of currency notes can occur especially during inadequate hygiene, adding cross-contamination from between currency notes and foodstuffs.

The association of E. coli, Klebsiella spp. and Staphylococci spp. of the current work would differ when emphasis is either given to different food vendors or Naira denominations, as respectively demonstrated in Figs. 2 and 3. When emphasis is given to the different food vendors (Fig. 2), higher E. coli and Staphylococci spp. loads were respectively found on Naira denominations recovered from the meat and fish sellers. This occurrence might reflect the nature of foodstuffs dealt by these vendors. High moisture content, blood, and intestinal components in both fish and meat samples have the capacity to provide a sufficient reservoir for various degrees of bacterial proliferation. When emphasis is given to the Naira denominations recovered from different food vendors (Fig. 3), the ₦100 and ₦200 obtained higher E. coli and Staphylococci spp. loads respectively, whereas ₦500 had higher Klebsiella spp. and Staphylococci spp. loads, with the least isolated bacterial contaminants at ₦10 and ₦5 (Klebsiella spp.), ₦5 (Staphylococci spp.), and ₦1,000 (E. coli). The increasing TVC trend across Naira denominations of ₦100, ₦200 and ₦500 in Table 2, can be seen to corroborate the data presented in Fig. 3, which are respectively loaded with E. coli, Klebsiella spp. and Staphylococci spp., attributable to the frequency of its usage in daily/various food market transactions. In addition, the ₦1,000 currency note appeared the least (bacterial) contaminated, probably because it is least used in Nigeria’s daily foodstuffs transactions. This same reason apply to ₦5, which also appeared the least (bacterial) contaminated, as shown in Table 2, which also corroborated the data presented in Fig. 3, consistent with the least Klebsiella spp. and Staphylococci spp., and a low E. coli count, after the ₦1,000 note. Previous researchers have shown that the improper handling of currency money by food vendors can transfer bacteria from currency notes to humans (Michaels, 2002; Lamichhane et al., 2009). For the reason that the higher denomination notes obtained greater bacterial contaminants, with the exception of ₦1,000, authors of current work opine that the Naira denominations might associate with the degree of (bacterial) contamination, resembling the argument Uneke & Ogbu (2007) has made on this matter.

The prevalence of E. coli, Klebsiella spp. and Staphylococci spp. on currency notes from different food vendors suggested the following trend: E. coli (87.5%) > Staphylococci spp. (85%) > Klebsiella spp. (72.5%). Different from the current study, the findings of Yazah, Yusuf & Agbo (2012) reported much less data frequency with different trend of S. aureus (22.5%) > E. coli (12.5%) > Klebsiella spp. (5%) on the Naira currency notes. This less data frequency of bacterial contaminants could be attributed to the location of their study, which was in the Northern part of Nigeria. Northern Nigeria is well known to have hotter weather temperatures compared to Southern parts. Comparing the above data of Yazah, Yusuf & Agbo (2012) with those of the current study would suggest the Naira currency notes in the Northern Nigeria to have less bacterial contamination/load compared to those at the Southern Nigeria, which is the location where the current study was performed. However, based on the different food vendors, the prevalence showed the following trend: vegetable sellers (91.7%) > meat sellers (87.5%) > fish sellers (87.5%) > fruit sellers (75%) > grain sellers (66.7%). Similar bacterial contamination observations involving the Nigerian currency notes recovered from fish, meat and vegetable sellers have been shown in previous studies (Ahmed et al., 2010; Barua et al., 2019). Applicable to other countries, the Nigerian currency notes when handled in an unhygienic manner would most likely supplement the frequency of bacterial contamination. This situation can arise from various sources such as atmosphere (air), body of handlers (hand, skin, wounds, etc.), counting machine, storage environment, soil, etc. (Awodi & Nock, 2001; Prasai, Yami & Joshi, 2008). Additionally, tongue-wetting of fingers appears a habit of many when counting money, which could serve as means of contaminating currency notes, fingers (Igumbor et al., 2007) as well as foodstuffs. For emphasis, the simultaneous handling activity between foodstuffs/food items and currency notes continues to serve as strong candidates capable of promoting as well as progressing foodborne disease incidence and spread through contamination and cross-contamination.

Conclusions

The bacterial contamination of Nigerian currency notes via a comparative study of different denomination notes recovered from local food handlers/vendors has been successfully investigated. Results showed about 81.7% of currency notes were contaminated with either E. coli, Klebsiella spp. or Staphylococcus spp., and in varying degrees. With the exception of ₦1,000 note, the (other) higher denominations of ₦500, ₦200, and ₦100 note recorded increased degree of contamination over the lower denominations of ₦50, ₦20, ₦10, and ₦5 note. Naira denominations note from meat, fish, and vegetable sellers obtained higher level of bacterial contaminants compared to those of the other food vendors.

Increased awareness and education among food vendors and ready-to-eat food sellers is warranted if possible cross-contamination between currency notes and foodstuff is to be mitigated. The Central Bank of Nigeria (applicable to central banks at other countries) would have to increase the robustness of the existing retrieval system, if the bacterial contamination and re-contamination of Naira denominations are to be reduced. This would specifically help to ensure that such highly used higher denominations of ₦500, ₦200, and ₦100 note, as reported in this current study, do not remain in the circulation process for too long. Additionally, this current study contributes to the call for increased awareness at the local, state and federal government levels in Nigeria, to place more emphasis on the potential public health risks that can potentially arise from the simultaneous handling of money and foodstuffs.

There is the possibility that nutrient agar used in the current study could be limiting the colony richness of some specific bacteria over others on the spread plate. Hence, in order to enhance the richness of colony on the spread plate, considering more bacterial specific agar for use is very needful in future studies involving same and or other currency notes versus similar (to current study) and or other food vendors. Another direction of future studies could be the use of 16s rRNA sequencing as well as matrix-assisted laser desorption and ionisation (MALDI), which are among promising molecular level microbiological techniques. These techniques can help increase the accuracy of bacterial identification when applied to studies investigating same (as well as other) currency notes versus same (as well as other) food vendors. It is also recommended that future studies could determine the prevalence of other microorganisms like yeast, fungi, and virus on currency notes across various food vendors, by comparing different locations.

Supplemental Information

Supplemental Information 1 Raw Data.

Click here for additional data file.

Additional Information and Declarations

Competing Interests

Author Contributions

Data Availability

Charles Odilichukwu R. Okpala is an Academic Editor of PeerJ.

Chigozie E. Ofoedu conceived and designed the experiments, analyzed the data, prepared figures and/or tables, and approved the final draft.

Jude O. Iwouno conceived and designed the experiments, authored or reviewed drafts of the paper, and approved the final draft.

Ijeoma M. Agunwah performed the experiments, analyzed the data, prepared figures and/or tables, and approved the final draft.

Perpetual Z. Obodoechi performed the experiments, prepared figures and/or tables, and approved the final draft.

Charles Odilichukwu R. Okpala analyzed the data, prepared figures and/or tables, authored or reviewed drafts of the paper, and approved the final draft.

Małgorzata Korzeniowska analyzed the data, prepared figures and/or tables, authored or reviewed drafts of the paper, and approved the final draft.

The following information was supplied regarding data availability:

Raw data is available in the Supplemental Files.

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
