# Peer review of "Bacterial contamination of Nigerian currency notes: A comparative analysis of different denominations recovered from local food vendors"

_PeerJ, doi:10.7717/peerj.10795_

## Round 0.1 · original submission · Major Revisions

Dear Dr. Ofoedu,

I agree with the issues raised by the two expert reviewers. Please revise your manuscript accordingly. I am looking forward to receiving your revised manuscript.

Best regards,
Elisabeth Grohmann

·

Basic reporting

1. The work presented in this manuscript is interesting, and relevant. The authors have determined the bacterial load on the Nigerian currency notes ₦1000, ₦500, ₦200, ₦100, ₦50, ₦20, ₦10, and ₦5 recovered from local fruit, vegetable, meat, fish, and grain sellers in Nigeria.

2. Please check the manuscript for grammatical errors.

Experimental design

1.The authors have mentioned that the samples were inoculated on nutrient agar plates and incubated at 37°C for 24 hours. Were these samples also inoculated on other media such as Brain Heart Infusion Agar, Tryptic Soy Agar, Blood Agar, or Mac Conkey’s Agar? Additionally, were the nutrient agar plates incubated at different temperatures and for different time-periods? Using two to four different media and different incubation conditions will increase the accuracy and reduce the bias as compared to using a single medium and incubation conditions.
2.The samples were prepared by immersing the notes in 100 mL Ringer’s solution, the beaker was shaken, the notes removed, and the solution was used as the sample for inoculation. Were these notes inoculated on the agar plates or observed microscopically after sample preparation? It would be interesting to know if the method used efficiently detaches all the bacteria from the surface of the notes or if additional steps are required for successful bacterial detachment.
3.The authors only mention using microbiological techniques for the identification of bacterial isolates. Was 16S rRNA sequencing or MALDI or a similar technique performed for the same? Using one of these molecular techniques in addition to the microbiological techniques is recommended for accurate identification of the bacterial isolates.

Validity of the findings

No comment.

Additional comments

1. Please check the mean values in Tables 2, 3, and 5.
2. It would be interesting if the authors add a short section comparing the results of the presented study with other similar studies.
3. Figures 2 and 3 contain an additional light orange legend, please clarify what it indicates.

Reviewer 2 ·

Basic reporting

Ofoedu et al, report about the degree of bacterial contamination and its dependency on the food vendor type and currency denomination(s). For this purpose, the authors have performed microbiology experiments. Similar information related to Nigerian currency and bacterial contamination has been published earlier, but the authors provided additional information about its relation to the food vendors. This information is useful, however, there are important questions that need to be addressed.

Experimental design

There are several aspects that require revision and/or clarification.
1. The authors have described, (line 163-164) 100 ml sterile ringer solution was used to immerse each Naira. It would be better if the composition of the ringer’s solution used in this study is also provided. Or details, if it was purchased?
2. When the authors provided duplicate values, are these values of the same samples (replicates), it’s not easy to understand from the method description.
3. The author should describe why only nutrient agar was used for the experiment?
4. I would like to appreciate the amount of work the authors did, characterize >250 samples for each fruit seller/each currency denomination. It’s a lot of work just to characterize the viable colony-forming unit by gram staining and growth media cultivation experiments.

Validity of the findings

1. The author mentioned the use of Mannitol salt and Eosin methylene blue media to characterize the isolates, however, there is no information presented in data, how many were pathogenic and non-pathogenic strains of staphylococcus.
2. Line-203: The TVC for the fruit seller is mentioned wrong, it should be 3X103-2.2X105, (according to the supplementary data provided).
3. I would like the authors to explain, if the TVC of ₦10 (2nd seller) is 1X103 then, how it is possible to have higher CFU/ml (1.1 for E.coli)?

Additional comments

1. The English language should be improved to ensure easy understanding for the international audience.
2. For eg. Line 160- The wire loops were sterilized by flaming, need to be removed and rewritten. The whole paragraph of preparation of materials /samples should be modified.
3. Some details are unnecessary, for eg. Line 174- “in an invert position”.
4. The authors need to cross-check all the values from all the tables, the data described in the table is confusing.
5. The time duration matters for bacterial transmission via currency note. Hence, the author should describe the time difference/duration between the sample collection and sample processing.
6. In the case of Meat, fish sellers, the sampling should have been done on blood agar. To provide rich nutrients for pathogenic strains that might be present.

---

## Round 0.2 · accepted · Accept

I agree with the two expert reviewers that your manuscript has considerably improved by the revision. It is now acceptable for publication in the hournal.

Congratulations!

·

Basic reporting

No comment.

Experimental design

No comment.

Validity of the findings

No comment.

Additional comments

The manuscript has definitely improved upon revision. The revised manuscript reads well with a clear methodology, discussion of obtained results, and conclusion.

Reviewer 2 ·

Basic reporting

Changes made.

Experimental design

Changes made.

Validity of the findings

Changes made.

Additional comments

The authors made all the suggested changes and significantly improved the manuscript. I approve the manuscript for publication.